# Heart Rate Variability and Accelerometry as Classification Tools for Monitoring Perceived Stress Levels—A Pilot Study on Firefighters

**DOI:** 10.3390/s20102834

**Published:** 2020-05-16

**Authors:** Michał Meina, Ewa Ratajczak, Maria Sadowska, Krzysztof Rykaczewski, Joanna Dreszer, Bibianna Bałaj, Stanisław Biedugnis, Wojciech Węgrzyński, Adam Krasuski

**Affiliations:** 1Faculty of Physics, Astronomy and Informatics, Department of Applied Informatics, Nicolaus Copernicus University in Toruń, Grudziądzka 5, 87-100 Torun, Poland; ewka@umk.pl; 2Faculty of Philosophy and Social Sciences, Institute of Psychology, Nicolaus Copernicus University in Toruń, Podmurna 74, 87-100 Torun, Poland; sadowska.maria88@gmail.com (M.S.); jdreszer@umk.pl (J.D.); bibianna@umk.pl (B.B.); 3Centre for Modern Interdisciplinary Technologies, Nicolaus Copernicus University in Toruń, Wileńska 4, 87-100 Torun, Poland; mozgun@mat.umk.pl; 4Institute of Safety Engineering, The Main School of Fire Service, Słowackiego 52/54, 01-629 Warsaw, Poland; sbiedugnis@sgsp.edu.pl; 5Fire Research Department, Building Research Institute (ITB), 00-611 Warsaw, Poland; w.wegrzynski@itb.pl

**Keywords:** stress, psychophysiology, monitoring, HRV, accelerometry, firefighters

## Abstract

Chronic stress is the main cause of health problems in high-risk jobs. Wearable sensors can become an ecologically valid method of stress level assessment in real-life applications. We sought to determine a non-invasive technique for objective stress monitoring. Data were collected from firefighters during 24-h shifts using sensor belts equipped with a dry-lead electrocardiograph (ECG) and a three-axial accelerometer. Levels of stress experienced during fire incidents were evaluated via a brief self-assessment questionnaire. Types of physical activity were distinguished basing on accelerometer readings, and heart rate variability (HRV) time series were segmented accordingly into corresponding fragments. Those segments were classified as stress/no-stress conditions. Receiver Operating Characteristic (ROC) analysis showed true positive classification as stress condition for 15% of incidents (while maintaining almost zero False Positive Rate), which parallels the amount of truly stressful incidents reported in the questionnaires. These results show a firm correspondence between the perceived stress level and physiological data. Psychophysiological measurements are reliable indicators of stress even in ecological settings and appear promising for chronic stress monitoring in high-risk jobs, such as firefighting.

## 1. Introduction

Firefighting is one of the most life-threatening, emotionally traumatic and stressful occupations. Firefighters routinely encounter stressors of various strength and intensity. They are exposed to severe acute and chronic stress, which can result in depression or occupational burnout [1]. Working in hazardous conditions causes exposure to a whole variety of extreme situations, such as witnessing death, suffering or participation in life-threatening events, perceived as a source of high stress that can result in post-traumatic stress disorder (PTSD) [2,3,4,5,6]. In addition to traumatic incidents, further sources of stress stem from daily working conditions, including organisational and administrative factors that reflect paramilitary, hierarchical, task- and tool-oriented, traditional power and command structures [7,8,9]. Furthermore, shift work system contributes to increased sleepiness and fatigue, and enhances the risk of injuries [10,11,12]. Chronic stress results in significant deterioration of health, frequent use of drugs and alcohol, and, in extreme cases, suicide, which is particularly related to symptoms of PTSD [13] and other psychiatric disorders [12,14]. Moreover, chronic stress causes an increased susceptibility to other stressors, which further leads to cardiovascular problems—the main cause of line-of-duty deaths among firefighters [15,16]. This reveals an appalling lack of continuous health monitoring and insufficient support of physical activity programs in high-risk jobs, such as firefighting service [17,18,19].

Among firefighters, chronic stress is the main cause of health problems, especially heart-related diseases however, these problems do not appear overnight. It takes months or even years of psychophysiological changes to develop a chronic condition and it is usually not possible to pin-point the exact event that triggered this shift. Additionally, the amount of exposure to acute stress necessary to prompt the chronic state depends on individual characteristics [20]. Therefore, only constant control of the pshychophysiological well-being in high-stress jobs can assure that the personnel receives the support they need exactly when they need it. This way it is possible to administer an appropriate mental health intervention or debriefing before it is too late. Real-time monitoring and on-line management of everyday stress experienced upon real-life events may be key to addressing this issue, as a preventive tool [15,17,18,21]. Unfortunately, control of the subjective, psychological perception of stress appears insufficient, as discrepancies were reported between self-assessed distress and objective evidence of harm [6,22]. Therefore, when examining stress, it is important to consider both its psychological and physiological nature. Regrettably, much of the stress-related research concerning firefighters was based on questionnaire studies, yielding an incomplete picture. Such procedures are retrospective and mostly carried out in laboratory settings [23,24]. Acute stressors observed in the laboratory rarely represent real-world situations accurately [25]. Moreover, only single-dimension characteristics of stressful events can be investigated in a laboratory setting, while real-life situations are far more complex [25]. Furthermore, self-report is not always reliable; a phenomenon reflected in overestimation of physical activity level upon self-evaluation [26,27]. Additionally, self-assessment is influenced by recall biases reflecting individual levels of coping mechanisms, experiences and even mood fluctuations [28]. Therefore, results obtained in laboratory settings via self-report may suffer from decreased accuracy and reliability, and thus cannot be generalised. For this reason it is important to consider ecological validity and physiological reactions to stressors in stress studies. It is necessary to investigate individual psychophysiological reactions exhibited upon experiencing real-life stress. Psychophysiological monitoring would allow a better understanding and management of stress dynamics and result in health-care consequences at individual and organisational level.

Development of wearable sensors designed to measure basic physiological parameters enable data collection in the course of daily activities and situations that may be relevant to an individual’s well-being and the ability to perform certain tasks [29,30,31,32]. Wearable devices, due to their mobility, high flexibility and connectivity, receive a great deal of attention from health-care environment [32,33]. Moreover, they offer an ecologically valid method for assessing stress levels with much higher accuracy than laboratory measurements, and open new possibilities for stress management and rehabilitation [29,32,34]. Several indicators of autonomic stress response can be recorded by biosensors; however, heart rate variability (HRV) is most commonly used. This parameter reflects the cardiovascular response to regulatory impulses affecting heart rhythm [35,36]. Therefore, HRV may serve as an indicator of the autonomic nervous system’s reaction to stressors [37,38], including job-related ones [34,39,40,41]. Reduced HRV is associated with deteriorated physical and psychological health, for example, decreased performance on cognitive and physical tasks [26,42,43,44], cardiovascular diseases [35], psychological stress [45], or depression and anxiety [46,47]. Real-time HRV measurement provides a suitable, unobtrusive and continuous method for detecting psychophysiological stress [33], and allows for development of health-care solutions for stress monitoring, management and rehabilitation. Moreover, HRV-based sensors can be useful as personal protection equipment for workers employed in high-risk jobs, operating in dangerous and stressful conditions, such as soldiers, firefighters, or policemen [40].

Despite its undeniable merits, stress assessment based solely on HRV measures is not accurate enough, since several different factors affect HRV for example, circadian rhythms, physical activities, body position [48]. Heart responses depend on the type, volume and intensity of exercise [49]. Additionally, it is difficult to fully describe the relationship between stress, different kinds of physical activity and HRV without the knowledge of the normal, daily fluctuations in HRV [50]. Unfortunately, one cannot rely on self-reports of physical activity, as it was proved that there is no correlation between the alleged activity and HRV fluctuations [50]. Therefore, it is necessary to employ an objective measure of physical activity. Combining HRV measurements with data from other sensors could aid the explanation of differences in HRV levels upon various psychophysiological states. Inclusion of accelerometric data helps to control the influence of physical activity by assessing the type of motion. Furthermore, it allows for delineation of epochs in the cardiovascular signal that correspond to different types of physical activity. In this way, it is possible to pool together HRV metrics calculated from natural, ecological short epochs of real-life signal recorded during similar episodes of physical activity. In a subsequent step, these separately analysed subsets of data can be compared with each other in order to differentiate stressful and non-stressful epochs. Such approach is far more sensible than analysing full 24-h recordings, which are highly non-homogeneous with respect to other factors influencing HRV. A full day may consists of several different situations, characterised by varying levels of stress, physical strain imposed on the body, and so forth. The standard approach to computing HRV indices in a sliding window does not allow for comparative analysis. Additionally, long signal recordings create certain difficulties in data analysis, mostly concerning data quality and problems with signal stationarity.

Few studies attempt to assess stress based purely on accelerometric data [51]. Changes in movement are not necessarily correlated with variations in levels of stress. This could be true in certain occupations however firefighters tend to be physically active not only during emergency calls, but also in their free time. Physiological parameters, such as HRV, skin conductance, breathing, reflect the psychophysiological state of the body far better. However, accelerometry makes it possible to determine specific behaviour patterns and overall activity level. Combined information from both types of signal provides a much more complete picture of a current bodily state. Typical accelerometric sensors measure body acceleration in three physical dimensions at high temporal resolution (usually 100 Hz). A combination of parameters measured along these three axes over a given period of time can be used to describe and distinguish specific components of movement that comprises a single behaviour or a series of activities [52]. Machine learning allows to classify these activities with a high degree of accuracy [53,54]. Classification is based on two accelerometric variables—static acceleration (gravity-related body orientation and position), and dynamic acceleration (induced by changes in movement) [52]. Therefore, accelerometer data alone suffice to infer activity levels and distinguish different types of physical activity, eliminating any need for direct observation or self-assessment. Integration of accelerometric measurement with HRV recordings in human stress monitoring appears to be an interesting possibility.

Little research has been done using similar methodology in the field of firefighting. A few studies attempted to assess stress levels based solely on HRV indices, both in laboratory [55] and ecological settings [17,56,57]. However, application of machine learning to classification of HRV-based assessment of physical, psychological and combined stress experienced by firefighters was performed only in laboratory conditions [58]. To our best knowledge, only one study applied artificial intelligence to analysis of ecological physiological recordings that consisted of combined HRV measurements and accelerometric activity [48]. The authors concluded that accelerometry significantly complemented HRV parameters, resulting in an increased stress prediction accuracy of 85.7%. This study was performed on a small sample size of unspecified workers involved in high-stress activities. However, firefighting is an extreme occupation, resulting in working conditions not easily matched by other stressful jobs. Relying on the outcomes of previous investigations and the availability of modern, portable technology for bio-signal registration, this pilot study aims to investigate methodological means to assess the level of stress experienced by firefighters in the field. Our hypothesis concerns the possibility to separate stressful incidents from non-incident situations by applying machine learning classification to HRV metrics collected in ecological settings from firefighters on duty. We further hypothesise that accelerometric measurements would further aid this process by providing a means to control for different types of physical activity recorded during 24-h shifts, encompassing both the participation in emergency incidents and leisure time. Moreover, the study attempts to classify epochs of the HRV-based biological signal according to the motion type and the level of stress experienced using various time domain and nonlinear HRV metrics in order to find the most reliable indicators of stress in the natural working environment.

## 2. Method

The ECG and accelerometer data were recorder from body-worn sensor during 24-h fire service shifts. Following every fire incident the firefighters filled in a short questionnaire, describing the time-frame and details of the incident as well as self-assessing the stress experienced upon action. Cluster analysis of the motion data differentiated the endpoints of the activities performed (motion clusters). Using the information about the time-frame of the incidents and the motion clusters, HRV analysis of the ECG time series served to evaluate the psychophysiological conditions. Taking into account two types of data allowed for automatic classification of data into more stressful (’incident’) and less stressful (’non-incident’) intervals.

### 2.1. Participants

Twenty six firefighters (all male, average age in years 22.4±1.8) serving in two fire departments were recruited to participate in this study: 12 cadets of The Main School of Fire Service in Warsaw, and 14 professionals working for the State Fire Service in Toruń. The subjects were familiarised with details of the experimental procedure applied in the study and signed an informed consent from. Furthermore, the subjects were informed about their right to withdraw from the study at any time without stating their reason for doing so. The study was approved by a local ethics committee.

Before being accepted into the State Fire Service of Poland, the candidates undergo extensive medical examinations. Due to very demanding working conditions, only completely healthy applicants are accepted into the service. Then, as firefighters, their health conditions are annually screened. The primary examinations regarding cardiovascular respiratory diseases are: resting ECG, blood pressure, cholesterol levels, spirometry, and chest x-ray (every 3 years). If any abnormalities are detected, the individual is withdrawn from the active service and directed to other duties, for example, office or training. None of the participants involved in this study has reported any intermissions in their active service.

### 2.2. Procedure

The participants’ psychophysiological reactions to work-related stress were recorded in ecological settings while the subjects remained on duty. Every morning, one of the researchers visited the fire station to equip two firefighters with fully charged sensor belts. Each participant was asked to wear it for their entire 24-h shift and not to remove it unless it caused serious discomfort. In case of an emergency incident, the subjects were asked to complete immediately after the event a short questionnaire concerning their subjective assessment of the hazards and risks encountered in action (perceived stress level). At the end of the 24-h shift, the researcher revisited the fire station to collect the questionnaires and sensor belts, and replace the latter with two charged ones for the next two participants. The collected equipment was recharged and refreshed, and data were downloaded from memory cards. The participants rarely took part in the experiment more than once, due to their work schedule. The procedure was repeated every day for the entire duration of the study.

The sensors made it possible to record the firefighters’ physiological activity during their normal day of work. The subjects were told not to change anything in their daily routine and behave as usually–work, play, rest and sleep. According to the participants’ reports, the hardware did not affect their ability to perform daily activities comfortably and safely (only one subject removed the belt during the tests due to comfort reasons). Fitting the proper size of the belt was most crucial in order to acquire low-noise signal. The sensor belt recorded the participants’ heart beat (used to calculate HRV metrics) and motion parameters of their performance (body acceleration measured in all 3 dimensions) during typical daily activities, as well as upon firefighter training’s and fire incidents.

### 2.3. Data Collection Instruments

#### 2.3.1. Stress Assessment

In order to assess the level of stress perceived by the participants during emergency incidents, a short questionnaire was developed specifically for this study. The questionnaires were completed immediately after returning from an emergency call as a form of debriefing. This was purposely intended to perform the evaluation of stress in a most ecological manner possible. The form consisted of eleven questions concerning the assessment of fire incidents in terms of threats, challenges, familiarity, effort, cooperation and degree to which the situation was stressful for the participant. Participants indicated the extent to which they agree or disagree with each statement on a 5-point *Likert-type* scale (ranging from ’strongly agree’ to ’strongly disagree’). Moreover, in an additional descriptive question, the subjects were asked to characterise the situation they had confronted, in a few short sentences, describing the type of action, equipment used, difficulties encountered and resulting victims of the incident. Last but not least, the participants were asked to state the approximate time-frame of the incident. Information on start- and end-points of the firefighting events made it possible to delineate high-stress periods in the firefighters’ daily schedules.

#### 2.3.2. Physiological Data Acquisition

Physiological data was collected with the Equivital™ EQ-02 Life Monitor sensor belts onto an internal SD card. ECG waveform was recorded at the resolution of 256 Hz and tri-axial acceleration was acquired at the sampling rate of 250 samples per second. Additionally, skin temperature and respiration rate were recorded. The sensor was mounted on an ergonomic belt (available in different sizes), worn around the chest and equipped with three dry ECG leads. The belt’s design allowed for an all-day data acquisition independent of the subject’s activity, even during physically demanding tasks. This equipment was previously used in studies involving firefighters [59] and soldiers [60]. Since the accelerometer is attached to the chest, its measurements are correlated with body movements and can serve to extract information about the subject’s behaviour, such as type of movement or position (e.g., running vs. walking, standing vs. lying).

### 2.4. Data Pre-Processing

The Equivital™ EQ-02 Life Monitor hardware includes proprietary pre-processing software that performs on-line ECG data filtering and derivation of secondary measures: heart rate and beat-to-beat (R-R) interval time series. Although the computation algorithm is not explicitly stated, the authors expect it to be the one described in Reference [61]. R-R series is formed by detection the so-called R-waves (i.e., peaks in voltage amplitude) in raw ECG reading and measuring the time between consecutive peaks (Figure 1).

### 2.5. Data Analysis

#### 2.5.1. Data Time Series Partitioning

Epochs (episodes) of different physical activities were determined based on similar accelerometric parameters. The accelerometer data was classified using cluster analysis (clustering), a data mining technique grouping a set of objects in such a way that the objects in one group (called a *cluster*) are more similar to each other than to the objects in the other groups. Clustering of time series data enforces the usage of indirect feature extraction methods. The most straightforward approach involves computing a *sliding window* over the time series (i.e., a fixed-size subset of consecutive elements of the time series) and composing a feature (or observation) vector using simple statistics that convey general information about a particular window. This approach is commonly used for human activity classification using accelerometer data [62,63]. Delineated clusters of motion data can be understood as episodes of different types of movements. For more details on the clusterisation method used see Appendix A.

Subsequently, the R-R time series was partitioned in two consecutive steps. (1) First, the accelerometric data partitioning was used to directly extract the corresponding segments of the R-R time series by aligning the endpoints of the R-R epochs with those of the clusters of motion data. In this way it was possible (a) to determine all R-R time series epochs recorded during similar, consistent conditions (one particular type of activity), and (b) to group the R-R series according to these conditions (movement types). Such analytical design allowed us to compare HRV metrics for all subjects divided into separate clusters, that is, during *different activities (physical effort)*. (2) Next, the R-R epochs created in the first step were divided further based on the reported time frames of the emergency incidents. This resulted in separation of the episodes of increased stress from the rest of the signal recorded. The second step enabled us to compare HRV metrics for all subjects *within* a given cluster between episodes characterised by different physiological load, that is, during (*similar activities (physical effort)*) under *different amount of stress* (’incident’ vs. ’non-incident’ condition).

#### 2.5.2. HRV Analysis

Heart rate variability analysis was performed using standard methods implemented in RHRV, an open-source package for R environment. HRV measures were calculated from epochs of the R-R time series, delineated in two subsequent steps described above (subsection ’Data time series partitioning’). HRV indices were calculated using linear methods in time domain (SDNN, SDANN, SDNNIDX, pNN50, SDSD, rMSSD, IRRR, MADRR, TINN, HRVi), and non-linear methods, such as Poincaré plot (SD1, SD2) and entropy measurements (Correlation Dimension, $D_{2}$, and Scaling exponent, ScalExp). For description and explanation of abbreviations of all the HRV parameters applied in this study see: Table 1. Time-domain analysis is well grounded in psychophysiological research, and the indices calculated reflect the functioning of the sympathetic and parasympathetic branches of the autonomic nervous system [64]. Physiological connotations of the non-linear dynamics of HRV are less well understood; nevertheless, the calculated measures are considered to be very sensitive in discriminating various psychophysiological states [65,66].

### 2.6. Statistical Analysis

Stress self-assessment questionnaire data were analysed in SPSS statistical software. Correlations between positions of the questionnaire were calculated as Spearman’s *rho* index. Factor analysis of the questionnaire answers was performed by the principal component method using the Oblimin oblique rotation of the axes (with Kaiser’s normalisation). Statistical significance was assumed at *p* < 0.05.

For the statistical analysis involved in machine learning and classification, Python packages *sklearn* and *seaborn* were used. Test data consisting of random R-R epochs were labelled as ’incident’ or ’non-incident’ upon binary classification with a Naíve Bayes classifier. The evaluation was based on ROC analysis performed on the averaged classification results, using stratified shuffle splitting with 300 folds (due to the highly unbalanced data).

Statistical analysis of physiological data was performed in the R software package. Due to the fact that emergency incidents constituted only a small part of the daily schedule, far less data were collected during ’incident’ than during ’non-incident’ activities. Moreover, it is very likely that certain types of activities are not present during incidents. Therefore, due to the disproportion in cluster sizes, mean values of HRV indices were compared between ’incident’ and ’non-incident’ subsets with a Welch’s *t*-test.

## 3. Results

### 3.1. Self-Assessment of Perceived Stress

A total of *N* = 43 emergency incidents were recorded (23 for cadets and 20 for professional firefighters). The answers indicated that in most cases the subjects experienced mild stress or no stress at all. Only a few emergency incidents were marked as stressful, while the grand majority was assessed as highly routine, typical interventions. Average answers and their standard deviations were summarised for each question and for the whole questionnaire in Table 2. The total score did not show normal distribution, exhibited a skew towards lower values (skewness 0.849) and a median of 22 points with a min-max range of 13–38 (in a theoretical range of 11–55 points). Information about the emergency incidents was retrieved from the open question. Quality analysis of the answers revealed that the majority of the emergency incidents recorded were grass fires, minor city incidents and training’s, and only a few were serious events, such as a burning building. Several statistically significant correlations were found between the positions of the questionnaire (Spearman’s *rho* between 0.31 and 0.79). The correlation matrix determinant equaled to 0.005, the Kaiser–Meyer–Olkin (KMO) index—to 0.6, and statistically significant results of the Bartlett’s test all indicate that our data are suitable for factor analysis. Principal component analysis with Oblimin rotation of the axes performed on the answers to the questions revealed 4 separate factors (loadings’ values of the questions loading each factor are presented in brackets): (1) ‘danger‘ (Q6 = 0.885, Q4 = 0.859, Q5 = 0.770, Q7 = −0.666, Q1 = 0.539); (2) ’satisfaction’ (Q10 = 0.866, Q11 = 0.833); (3) ‘involvement’ (Q9 = 0.926, Q8 = 0.828); (4) ‘routine’ (Q3 = 0.797, Q2 = 0.736). The model explained 72.8% of variance. Reliability of the questionnaire was controlled by calculation of Cronbach’s alpha, which was equal to 0.71 for the whole scale.

### 3.2. Physiological Recordings

A total of 26 sets of 24 h recordings of physiological data were gathered, and following initial data quality inspection, 25 sets were accepted for further analysis. The whole data set consisted of 485 h of recordings, out of which 45 h 15 min were recorded during 39 fire incidents. 17 subjects were involved in some kind of incident (9 cadets and 8 professionals). The attended fire incidents were reported by the firefighters to had lasted varying amounts of time (17% was < 18 min, 52% < 43 min, 87% < 85 min). One incident was annotated as 6 h long.

### 3.3. Clusters of Acceleration Data

As an unsupervised learning method cluster analysis allowed to delineate and group together episodes of similar accelerometric activity, however, without labelling them. Figure 2 shows an exemplary recording from one of the participants, separated into 84 episodes grouped into 20 clusters. During his shift, the firefighter participated in four emergency incidents marked by grey boxes.

The immediate observation from Figure 2 is that the peeks of acceleration variance are connected with different clusters, which can be understood as high intensity and low-intensity movements. Around 11 p.m. we can observe long episodes of resting, where only few clusters are depicted (no. 3, 7, 8), which are not present during day and correspond to different horizontal positions of the subject. It is very interesting to note that the respiratory and heart rates increase during incidents, as well as during normal activities. Around 10 p.m. we see a sudden drop of the temperature connected with temporary sensor dismounting due to showering.

Prior to comparing HRV indices between ’incident’ and ’non-incident’ clusters, signal partitioning was controlled for incorrect classification *indirectly* based on the HRV metrics; a situation, when clustering is influenced by its own methodology and therefore not reliable. We identified two theoretical instances when such a problem could occur: (1) when the incident is present in only one or two clusters, and (2) when epoch length affects HRV metrics. The former situation indicates that classification of R-R time series episodes as ’incident’ is based on the presence of a very specific type of activity, occurring *only* during incidents. However, in our study 4 clusters were found to contain both ’incident’ and ’non-incident’ episodes. To investigate the latter problem of episode length influencing the HRV values, we calculated correlations between episode length and HRV values.The highest correlation Spearman rank correlation observed in our study was 0.22 (p<0.01) for SDANN. Despite the fact, that this correlation was weak, the SDANN metric was considered possibly affected by episode length and therefore rejected from further analysis. Logistic regression confirmed that the cluster number did not predict attribution to an emergency event (’incident’). This allows to assume that clustering alone did not carry significant information about the incidents. Some clusters did not contain any data from emergency incidents, however, that is easily explainable by the specificity of motion represented by a given cluster. For example, certain clusters contained large fractions of data where the subject remained in a stationary, horizontal position (most probably sleeping), that should not (and hopefully did not!) occur during the incident. In the clusters reflecting an emergency event, data proportion of ’incident’ to ’non-incident’ recordings was consistent with the overall data.

### 3.4. Classification of R-R Time Series Episodes

In 6 out of 10 clusters, less than 2 ’incident’ events were present, and therefore were excluded from further analysis. In the remaining 4 clusters, ’incident’ events constituted on average 8% of all episodes. Differences in HRV metric between the ’incident’ and ’non-incident’ conditions were analysed by pooling together all epochs of each type and comparing them with a Welch’s *t*-test. The results revealed statistically significant differences for several HRV metrics. Considering further partitioning into episodes varying by physical activity, HRV parameters differed between the ’incident’ and ’non-incident’ conditions in the overall dataset, but not necessarily within individual episodes. Standard deviations of the HRV metrics within the final partitioning are high; however, in most cases a tendency towards lower mean HRV upon ’incident’ can be noted. Table 3 summarises the results of the location test performed on various HRV metrics, along with the corresponding mean values and standard deviations.

Average values of all HRV indices were compared between clusters with a one-way ANOVA, showing significant differences (p<0.001) for SDANN, IRRR, TINN, HRVi, SD2, and CD. Post-hoc pairwise analysis revealed that most significant differences could be found for clusters 2 vs. 6, and 1 vs. 2, accordingly.

Best results of individual R-R episode classification as ’incident’ or ’non-incident’ was achieved with AUC score of 0.7 (0.65 for a feature set without information on cluster number). More granular clustering (higher k-parameter) decreased the classification accuracy. Ranking the HRV metrics by Information Gain slightly improved the classification score. Final feature set composed of the following metrics, that were most sensitive in detection of stress: ClusterID, ScalExp, pNN50, MADRR, IRRR, HRVi, CD. Figure 3 summarises the classification results.

## 4. Discussion

### 4.1. Outcomes and Applications

This study aimed to detect stressful events experienced by firefighters on duty. Accelerometric motion data and corresponding physiological ECG signal were collected throughout a 24-h shift and analysed with techniques based on machine learning. Psychophysiological stress was assessed in the form of HRV paramenters calculated form the ECG-based R-R time series. The R-R series were subjected to automatic binary classification into episodes recorded *during* or *outside of* a potentially stressful firefighting event (’incident’). It was possible to distinguish the ’incident’ from ’non-incident’ episodes based on respective HRV parameters. However, separation results were better when types of movement were taken into consideration. Episodes representing different types of physical activity corresponded to motion clusters of accelerometric time series, based on motion parameter vectors constructed with accelerometric data (Figure 3). Therefore, each epoch of the R-R series analysed was (1) tagged as ‘incident’ (assumed *increased stress* levels) or ‘non-incident’ (assumed *baseline stress* levels), and (2) assigned to a certain type of physical activity.

There is a lot of convincing empirical evidence showing that due to its biological mechanisms, stress, especially chronic and acute, has a negative effect on various systems of our body, primarily immune, digestive, cardiovascular and hormonal. Therefore, it is important to determine the psychophysiological mechanisms that link stressful situations and their health consequences. Analysis of heart rate variability is a non-invasive, quantitative measurement of the activity of the autonomic nervous system of the heart, allowing to assess its adaptability [67]. In the state of health, this allows to respond quickly to changes in homeostasis and to modify heart function accordingly, depending on changing conditions. Stress is associated with changes in autonomic activity that disrupts normal homeostatic processes. Physiological changes caused by stress are reflected by changes in HRV. Excessive emotional tension negatively affects HRV. Thus, low HRV levels are associated with disorders of regulatory and homeostatic functions of the autonomic nervous system (ANS), which reduces the ability to deal with internal and external stressors [67]. Therefore, HRV is an electrocardiographic method that allows measurement of ANS responses in various situations, for example, when experiencing mental stress at work, and to observe changes that occur at the level of physiological mechanisms on an ongoing basis, making health prevention possible. The clinical value of HRV as a reliable indicator of stress and a predictor of various health problems is further supported by the fact that it is a non-invasive and simple diagnostic method.

Not all motion clusters were present during an emergency situation. This is understandable when considering types of physical activity, such as sleeping/lying motionless, which, for obvious reasons, do not appear during incidents. Four out of ten motion clusters constituted of sufficient ’incident’ episodes to compare various HRV parameters between the ’incident’ from ’non-incident’ conditions (Table 3). For each motion cluster only one or two HRV indices differed significantly between conditions, probably due to high variance and unequal lengths and numbers of epochs within each condition. Fire emergencies (’incidents’) contributed to a small percentage of the firefighters’ daily activities. Therefore, the ‘incident’ sample was underrepresented with respect to ‘non-incident’ data, which contained many more episodes. Moreover, other HRV metrics distinguished the two conditions within each of the analysed clusters. Statistically significant differences were found in SD2 and D2 within cluster 1, IRRR and SDNNIDX within cluster 6, and pNN50, SDSD, HRVi, SD2 and ScalExp within cluster 8. This might be explained by different general sensitivity to HRV change found among the various indices known to reflect different aspects of the autonomic nervous system activity [65,68]. Similarly, distinct HRV metrics may react differently to changes in cardiovascular activity caused by a particular type of motion (represented by a given motion cluster), and/or may present different sensitivity to stress-related changes in HRV (’incident’ and ’non-incident’ epochs within that motion cluster).

The general trend showed, as expected, lower HRV values upon stressful situations. Previous studies recommend the use of HRV measurements as a psychophysiological marker of stress [69]. Studying stress in a natural environment is definitely more ecologically accurate, however, it leaves no control over exact timing and amount of stress experienced by the subjects. Therefore, differences in mean HRV values between the more and less stressful conditions exhibit lower statistical significance. Nevertheless, our results correspond to the outcomes obtained in artificial laboratory settings. In this study we confirmed that HRV parameters are valid psychophysiological indicators that can distinguish stressful situations beyond the laboratory—in ecological settings.

Subjective stress levels experienced by the firefighters were estimated by a short questionnaire invented for the purpose of this study. Although dedicated standardised psychometric tests do exists, we decided to introduce a new questionnaire for three reasons—(1) in order to assess the stress experienced upon *each* incident, and (2) to avoid imprecise assessment based on imperfect recollections [70] and obtain accurate answers to self-satisfaction questions, the firefighters had to undertake the questionnaire *directly* after the event, while still on duty and therefore it needed to be as short as possible; (3) a standardised psychometric test should be administered under professional supervision of a psychologist, which was impossible is ecological settings. Moreover, psychometric data analysis revealed good reliability of our questionnaire (Cronbach’s alpha equal to 0.71). Factor analysis suggested the existence of four sub-scales within the questionnaire, assessing different aspects of a stressful emergency event and the corresponding actions of the firefighters. The sub-scales evaluate the danger imposed by the incident, level of routine of the intervention as well as degrees of personal involvement in the firefighting action and satisfaction experienced from the operation.

ROC curve analysis showed that appropriate decision criteria allowed to classify 15% of incidents with no false alarms, which may seem to be a huge underestimation. However, questionnaire results of stress level self-assessment indicated that only a few emergency actions had been evaluated by the participants as moderately stressful. Only in 14% of the reports (6 out of 43) the answer to Question 1. ‘To what degree was the situation stressful to you?’ was rated as 3 or more on a 1-5 *Likert-type* scale. The 15% of incidents classified as *true positive* may represent the *actual* amount of stressful incidents. These results suggest a good correspondence between the psychometric and physiological data, further increasing the reliability of psychophysiological parameters as markers of stress.

In personal communication, the firefighters reported experiencing the strongest agitation usually before the beginning of an emergency intervention. The moment of receiving dispatch instructions and departing for action was indicated as very stressful, while once the situation recognition is over, routine action and typical behaviour allows for good control and planning, resulting in decreased stress. Following an emergency incident, firefighters report low stress and life threat upon action. Therefore, the initial psychophysiological activation is taken into account *neither* in the post-action debriefing *nor* in the self-assessment questionnaires. It appears that self-reporting of stress remains non-discriminative and non-informative in this matter. Therefore, the nature of both chronic and acute everyday stress experienced by the firefighters may remain neglected, despite the fact that it is inherent to the profession. Strong, acute stressors occur with various frequency; however, chronic stress accompanies high-risk professionals throughout their career on a daily basis. The inability to control and predict the outcome of a difficult and/or life-threatening situation results in a frequent mobilisation of the sympathetic branch of the autonomic nervous system. Experiencing chronic stress leaves insufficient time for the system to return to a balanced state, leading to the development of affective and other stress-related disorders. Moreover, the majority of firefighters experience very scarce psychological care, limited to an annual health screening, performed in a form of a survey or basic neurological examination [71,72]. Professional psychologists visit the fire stations rather occasionally, usually following acute traumatic incidents, while the issue of chronic stress remains unaddressed. Despite their lack of psychological background, firefighters need to recognise themselves any worrying symptoms observed in their peers. Alarming in itself, this problem is predicted to increase with the *smart-phone generation* joining the ranks of fire service. As this age group is characterised by lower levels of empathy, the recognition of chronic stress symptoms in firefighters and proper intervention will become even poorer [73]. It is therefore crucial to monitor levels of stress experienced by employees, especially in high-risk jobs, such as the firefighting service. Physiological monitoring may be very informative in this matter, providing objective, yet personalised and very detailed information about the daily profile of experienced stress.

The presented findings also bear other practical implementations. Since the same motion clusters were present in data profiles of different subject, motion partitioning can be considered universal and independent of individual characteristics of movement styles and HRV. In theory, it would be possible to assign tags to each delineated cluster, naming the corresponding motion. Therefore, it seems possible to construct a universal classifier, that could detect the type of motion on-line, name it, and assess the current level of stress experienced during a given activity. This type of application would be based on easily collectable data, retrievable with noninvasive wearable sensors even upon movement. Furthermore, based on individual profiles of HRV parameters and motion characteristics, it is possible to construct classifiers distinguishing between individuals. On-line data collection, processing and classification would aid in application of the concept of *dynamic analytical risk assessment* at the emergency scene [74,75]. The Incident Commander, supported by a computer system, is responsible for emergency scene risk management, based on continuous monitoring of the firefighters’ life parameters. Dynamic reactions to predefined patterns may help to act accurately and without delay in life-threatening situations. This approach resembles the tactics currently used by some fire services, such as the German Threat Matrix [76,77] or the British Generic Risk Assessment [78]. Computerised risk monitoring is more reliable and allows the Incident Commander to concentrate on other (no less demanding) aspects of response leading. Automatisation of risk assessment appears to be a very promising technology with rapidly increasing applications and wide-spread use in the future.

### 4.2. Limitations & Outlook

A possible limitation of every study carried out in ecological conditions is data quality. ECG recordings may contain substantial noise inherent to using dry sensors and movement artifacts. However, a proper fit of the sensor belt allows for acquisition of high-quality, clean data even in natural settings [79]. Hence, in order to analyse reliable and good quality data, the firefighters were instructed how to wear the sensor belts properly, while data quality was hand-inspected, poor-quality signal was rejected, and the R-R data was partially interpolated when necessary. Similarly, a lack of baseline HRV levels recorded prior to the firefighting shifts could be a potential drawback. However, during every 24-h shift several routine activities were performed, which can be safely treated as baseline or internal control. Nevertheless, baseline measurements provide certain benefits and should be implemented in future studies.

Another potential confinement originates from the assumption that no stress was experienced outside of emergency incidents, while each incident was stressful. This might not necessarily be true, and result in decreased ’incident’/’non-incident’ classification accuracy. A possible solution might involve further clusterisation of the motion clusters regardless of emergency incidents, in an attempt to distinguish high and low stress based on acceleration and HRV signals only. Unfortunately, too few incidents were recorded to satisfy the statistical requirements of such an analysis.

Moreover, the heavy firefighter gear worn in action might influence accelerometric and HRV measurements by imposing an additional load on the body, obstructing motion and imposing strain on the cardiovascular system. As a result, the same type of motion could fall into different clusters depending on the body load. Presence of the same clusters in both ‘incident’ and ‘non-incident’ conditions indicate this is probably not the case, nevertheless, fewer types of physical activity are present during the latter condition. These matters should be addressed in future studies, analysing data from exercises in full gear, stressful social or other events experienced on duty, but outside of emergency incidents.

Furthermore, restricted generalisation of the results stems from the small sample size and high homogeneity of the group (all young healthy Caucasian male subjects). Factors such as race, age, sex, and physical fitness strongly influence HRV values [80,81,82,83], while females exhibit different changes in HRV profiles in response to physical exercise [84,85] and stress [86,87]. Construction of a universal classifier for motion type and experienced stress would require the learning data set to consist of a large number of samples collected from a very differentiated group of subjects. Alternatively, better accuracy could be achieved by constructing age-stratified and race- and sex-specific classifiers, based on profiles of biometric parameters from the general population. Moreover, inclusion of concurrent video recordings into future studies could result in tagging (naming) the delineated types of physical activity, represented by particular clusters.

Finally, focusing on traumatic events during stress monitoring in high-risk jobs [6,88,89] may not be enough, as chronic stressors continue to influence the employees in a personal fashion [90]. Future studies should take into account individual differences in reactions to traumatic but also routine events. Moreover, detailed information received prior to a stressful event may turn out crucial for limiting chronic stress attributed to preparation for an unknown incident. Studies on this matter may reveal the need to improve the dispatch procedure and increase the amount of information available prior to emergency events in high-stress jobs, such as firefighters, policemen, doctors, paramedics, and many more.

## 5. Conclusions

Application of machine learning techniques to physiological and motion data allowed to create an automatic classifier of events. Based on accelerometric measurement it was possible to delineate different types of motion. Analysis of HRV parameters calculated from corresponding R-R series of ECG recording allowed for further recognition of stress levels. Classification accuracy of stressful events paralleled the incidence of actions reported as stressful in a self-assessment psychometric questionnaire. Psychophysiological measurements appear to be reliable a biomarker for monitoring chronic stress, especially in high-risk jobs. Considering the low cost and high availability of good quality sensors, it is feasible to collect data on-line in ecological environment. In highly stressful occupations, such as firefighting, real-time psychophysiological monitoring may be treated as early screening for symptoms of chronic stress, and basis for further contact with professional care. Such technology is still incipient, nevertheless, it is necessary to open a discussion about this problem between fire safety researchers, firefighters, engineers and design professionals. Moreover, automatic detection equipment could significantly improve stress monitoring not only in high-risk jobs, such as firefighting, but also other professions exposed to everyday stress. Proper evaluation of chronic occupational stress would improve prevention of stress-related disorders, enhance job satisfaction and quality of life of the employees, as well as decrease errors inflicted by the human factor.

## Figures and Tables

**Figure 1 sensors-20-02834-f001:**
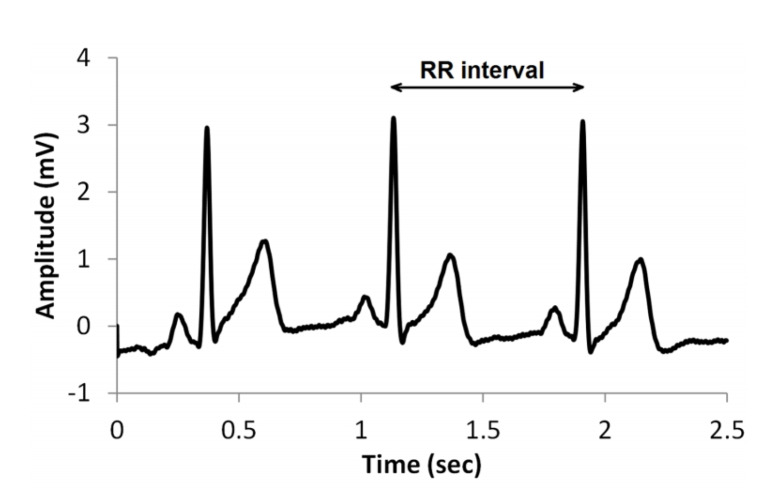
A typical electrocardiograph (ECG) signal showing the RR interval.

**Figure 2 sensors-20-02834-f002:**
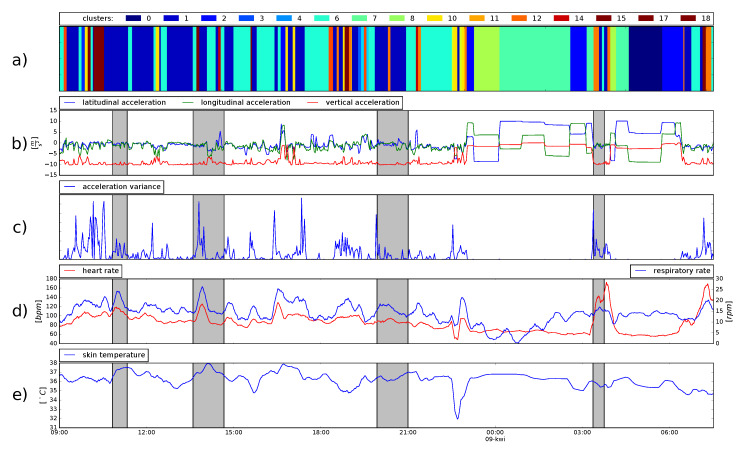
Exemplary data from a 24 h recording retrieved from one of the participants. Consecutive graphs (marked by subsequent letters of the alphabet) correspond to the same timeline. (**a**) depicts clusters of motion time series. Motion data consists of three-axial acceleration measurements, and acceleration variance, presented in (**b**,**c**) respectively. (**d**) illustrates changes in HRV (red line, left axis) and respiration rate (blue line, right axis), while (**e**) shows fluctuations in body temperature.

**Figure 3 sensors-20-02834-f003:**
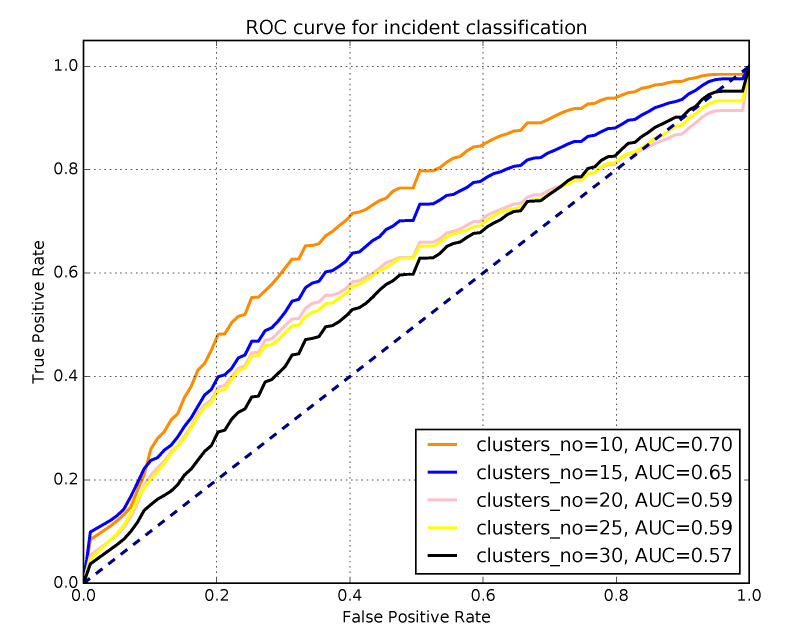
Classification for ’incident’ detection based on HRV parameters, performed on 5 different sets of R-R epochs delineated via accelerometer data using 5 different clustering parameters (k=10,15,20,25 and 30).

**Table 1 sensors-20-02834-t001:** Description and explanation of abbreviations of the heart rate variability (HRV) parameters applied in this study. Hereafter, NN refers to the time elapsed between consecutive ’normal-to-normal’ beats of the heart, equivalent to R-R, that is, peak-to-peak period between the R-waves on the ECG.

HRV Index	Description
SDNN	Standard deviation (SD) of the NN (R-R) inter-beat intervals
SDANN	Standard deviation of averaged over 5-minute periods NN (R-R) intervals
SDNNIDX	Mean value index (IDX) of SDNN
pNN50	Proportion of the adjacent (successive) NN (R-R) intervals greater than 50 ms
SDSD	Standard deviation of the successive differences between the adjacent NN (RR) intervals
rMSSD	Root mean square differences between the successive NN (R-R) intervals
IRRR	Length of the interval between the first and the third quantile of the ΔRR time series
MADRR	Median of the absolute values of the successive differences between the adjacent NN (R-R) intervals
TINN	Triangular interpolation of the NN (R-R) interval histogram.
HRVi	Index reflecting the slowing down of the heart
SD1	Dispersion of the points along the minor axis of the Pointcare plot (SD of the short-term R-R interval variability)
SD2	Dispersion of the points along the major axis of the Pointcare plot (SD of the long-term R-R interval variability)
$D_{2}$, ScalExp	Correlation dimension and scaling exponent alpha, non-linear dynamics measures of time series.

**Table 2 sensors-20-02834-t002:** Results of self-assessment of stress experienced upon emergency events. Mean score and SD calculated for each position of the questionnaire and for the total score.

No.	Question ^a^	Mean	SD
1.	To what degree was the situation stressful to you?	1.91	0.75
2.	To what degree was the situation not a challenge to you?	3.56	1.03
3.	To what degree was the action routine?	3.60	1.20
4.	To what degree was the situation a threat to your life?	1.47	0.74
5.	Did the situation endanger civilians in your surroundings?	1.84	1.15
6.	To what degree did the situation endanger other firefighters involved in the action?	1.44	0.73
7.	To what degree was the situation not a threat?	3.77	1.25
8.	Assess the amount of effort you had to undertake in this situation	2.16	0.92
9.	Was your involvement crucial to the action?	2.81	1.16
10.	Assess how satisfied you are with your actions during the incident	3.77	0.81
11.	Assess how satisfied you are with your co-operation with other participants throughout the action	4.12	0.66
12.	In a few short sentences, characterise the situation and your feelings about it (state the type of action, equipment used, difficulties encountered in action, victims/injured parties [people, animals and possessions], and any information that seems important to you) ^b^		
	Total score	22.81	5.36

^a^ Rated on 5-point Likert-type scale: ranging from strongly disagree (1) to strongly agree (5) ^b^ Open question.

**Table 3 sensors-20-02834-t003:** Results of Welch’s *t*-test significance and values of compared means (with confidence intervals) for ’incident’ vs. ’non-incident’ episodes calculated for various HRV parameters within motion clusters that contained a sufficient number of ’incident’ events. Asterisks indicate statistically significant results: * *p* < 0.05; ** *p* < 0.01.

	ClusterID (No. of Incident/Non-Incident Episodes)			
	1 (25/166)	2 (3/48)	6 (11/151)	8 (5/126)
SDNN (ms)	*p* = 0.312	*p* = 0.608	*p* = 0.203	*p* = 0.271
incident	117.92 ± 41.52	83.47 ± 35.30	112.12 ± 50.22	98.96 ± 27.27
non-incident	128.61 ± 65.50	107.45 ± 48.97	134.62 ± 64.33	27.27 ± 56.94
SDNNIDX (ms)	*p* = 0.166	*p* = 0.522	*p* = 0.050 *	*p* = 0.083
incident	93.52 ± 46.06	74.73 ± 43.20	99.24 ± 50.84	76.68 ± 19.02
non-incident	104.52 ± 59.65	95.65 ± 47.51	110.80 ± 60.70	19.02 ± 54.30
pNN50 (%)	*p* = 0.225	*p* = 0.735	*p* = 0.084	*p* = 0.018 *
incident	18.61 ± 13.89	11.88 ± 11.45	15.92 ± 14.29	13.99 ± 9.53
non-incident	23.30 ± 16.05	22.21 ± 14.17	26.36 ± 16.81	9.53 ± 16.58
SDSD (ms)	*p* = 0.225	*p* = 0.735	*p* = 0.084	*p* = 0.018 *
incident	65.00 ± 44.10	61.43 ± 47.03	60.93 ± 47.39	44.16 ± 16.89
non-incident	78.71 ± 58.71	81.26 ± 47.61	92.33 ± 61.78	16.89 ± 48.14
rMSSD (ms)	*p* = 0.308	*p* = 0.471	*p* = 0.447	*p* = 0.106
incident	64.98 ± 44.08	61.41 ± 47.01	60.92 ± 47.37	44.15 ± 16.88
non-incident	78.65 ± 58.54	81.17 ± 47.55	92.22 ± 61.52	16.88 ± 48.12
IRRR (ms)	*p* = 0.173	*p* = 0.607	*p* = 0.020 *	*p* = 0.069
incident	151.52 ± 63.06	80.00 ± 37.00	154.00 ± 97.90	115.00 ± 18.11
non-incident	168.31 ± 110.14	120.23 ± 74.10	177.93 ± 117.43	18.11 ± 85.11
MADRR (ms)	*p* = 0.991	*p* = 0.793	*p* = 0.922	*p* = 0.882
incident	20.64 ± 12.09	14.00 ± 10.00	16.27 ± 11.28	18.00 ± 8.17
non-incident	25.16 ± 17.99	20.84 ± 12.91	26.52 ± 16.03	8.17 ± 18.49
TINN (ms)	*p* = 0.991	*p* = 0.793	*p* = 0.922	*p* = 0.882
incident	335.49 ± 98.45	214.78 ± 102.79	315.20 ± 80.28	311.77 ± 23.04
non-incident	333.28 ± 136.37	243.49 ± 105.54	306.71 ± 105.84	23.04 ± 144.73
HRVi	*p* = 0.225	*p* = 0.735	*p* = 0.084	*p* = 0.018 *
incident	21.47 ± 6.30	13.75 ± 6.58	20.17 ± 5.14	19.95 ± 1.47
non-incident	21.33 ± 8.73	15.58 ± 6.75	19.63 ± 6.77	1.47 ± 9.26
SD1 (ms)	*p* = 0.331	*p* = 0.576	*p* = 0.246	*p* = 0.367
incident	45.96 ± 31.18	43.43 ± 33.25	43.09 ± 33.51	31.23 ± 11.94
non-incident	55.65 ± 41.51	57.46 ± 33.66	65.29 ± 43.68	11.94 ± 34.04
SD2 (ms)	*p* = 0.008 **	*p* = 0.648	*p* = 0.121	*p* = 0.021 *
incident	159.42 ± 52.28	108.34 ± 41.15	151.51 ± 65.12	136.26 ± 37.24
non-incident	172.22 ± 84.50	139.49 ± 62.37	177.44 ± 82.81	37.24 ± 74.53
$D_{2}$	*p* = 0.010 **	*p* = 0.400	*p* = 0.095	*p* = 0.296
incident	1.44 ± 0.06	1.25 ± 0.13	1.43 ± 0.04	1.46 ± 0.02
non-incident	1.35 ± 0.33	1.28 ± 0.35	1.39 ± 0.21	0.02 ± 0.15
ScalExp	*p* = 0.157	*p* = 0.767	*p* = 0.053	*p* = 0.015 *
incident	1.01 ± 0.11	1.19 ± 0.22	1.07 ± 0.20	1.05 ± 0.19
non-incident	0.91 ± 0.21	0.86 ± 0.29	0.92 ± 0.25	0.19 ± 0.23

## Data Availability

Please contact the authors with data requests.

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
