# Peer review of "Heart Rate Variability and Accelerometry as Classification Tools for Monitoring Perceived Stress Levels—A Pilot Study on Firefighters"

_sensors, 2020, doi:10.3390/s20102834_

Round 1

Reviewer 1 Report

This the very original scientific study provided with the goal of investigation of impact of stress episodes in the work shifts of firefighters on HRV. HRT is used to be the universal method to estimated different physiological and pathological reactions. The authors used also acetometer along with ECG and tested with created questionnaire. At the end it was shown that link between physical stress and physiological parameters (HRV). However there some suggestions to the study from my point of view. They should be considered or should be reasoned why they could not be corrected.

Thanks for providing very interesting and unusual study about interaction between physiological body reactions and stress during the stress at shift in firefighters.

The introduction, methods, results, discussion are well described. However I have some questions and corrections. You may of course base if you are not agree with me.

  1. Please provide the table with more detailed characteristics of the study group. The existence of some cardiovascular, respiratory and etc. diseases may impact the HRV data.
  2. Please provide the clear design in the scheme in the methods.
  3. Is there is the sense to divide the group into professionals and cadets. This way you need to compare the groups and show HRV difference between them.
  4. How can you validate and standardize the questionnaire? You say it was created by yourself.
  5. I do not see the analysis which the HRV parameters are mostly sensitive which are attributed to the physical stress.
  6. Also, I did not understand if you measured initial HRV without the shift (as a control group).
  7. Taking into the account that HRV can predict such pathological events such as sudden deaths, myocardial infarction even in young persons the revealing of extreme overload estimated by HRV maybe of big use in the future. You describe it in the discussion.

Author Response

We would like to thank the reviewers very much for their effort in the reviewing process and the valuable comments and remarks. These comments and remarks have helped us for further improvement of the manuscript. Please find below our answers for specific remarks. 

Reviewer 1

  1. Please provide the table with more detailed characteristics of the study group. The existence of some cardiovascular, respiratory and etc. diseases may impact the HRV data.

Before being accepted into the State Fire Service of Poland, candidates for firemen undergo extensive medical examinations. Due to the very demanding working conditions, only completely healthy people are accepted into the service. Then, as firefighters, their health conditions are annually screened. The primary examinations regarding cardiovascular respiratory diseases are: resting ECG, blood pressure, the level of cholesterol, spirometry, and the chest x-ray (every 3 years). If any anomalies are detected, the individual is withdrawn from the active service and directed to other duties, e.g. office or training. None of our participants has reported any breaks at their active service. 

Lines 151-157

2. Please provide the clear design in the scheme in the methods.

The scheme of the method section was added between section 2. and 2.1 that shows the rationale in the following method section. 

Lines 135-143

3. Is there sense to divide the group into professionals and cadets. This way you need to compare the groups and show HRV difference between them.

We agree with the Reviewer, that the comparison between professionals and cadets does not contribute to the results of our study and therefore we decided to remove related analysis from the manuscript.

4. How can you validate and standardize the questionnaire? You say it was created by yourself.

We explain our position briefly in the Discussion section:

“Subjective stress levels experienced by the firefighters were estimated by a short questionnaire invented for the purpose of this study. Although dedicated standardised psychometric tests do exists, we decided to introduce a new questionnaire for three reasons: (1) in order to assess the stress experienced upon each incident, and (2) to avoid imprecise assessment based on imperfect recollections (Lazarus 1984) and obtain accurate answers to self-satisfaction questions, the firefighters had to undertake the questionnaire \textit{directly} after the event, while still on duty and therefore it needed to be as short as possible; (3) a standardised psychometric test should be administered under professional supervision of a psychologist, which was impossible is ecological settings. Moreover, psychometric data analysis revealed good reliability of our questionnaire (Cronbach's alpha equal to 0.71). Factor analysis suggested the existence of four sub-scales within the questionnaire, assessing different aspects of a stressful emergency event and the corresponding actions of the firefighters. The sub-scales evaluate the danger imposed by the incident, level of routine of the intervention as well as degrees of personal involvement in the firefighting action and satisfaction experienced from the operation.“

Lines 403-415

5. I do not see the analysis which the HRV parameters are mostly sensitive which are attributed to the physical stress.

In order to make our message more clearer we elaborated on the following paragraph: 

“Final feature set composed of the following metrics, that were most sensitive in detection of stress: ClusterID, ScalExp, pNN50, MADRR, IRRR, HRVi, CD.”

Lines 344-346

6. Also, I did not understand if you measured the initial HRV without the shift (as a control group).

We did not measure baseline HRV prior to the firefighting shifts. However, during every 24-hour shift several routine activities (eating, sleeping, resting, free time activities) were performed, which can be safely treated as baseline or internal control. Therefore, exploiting  the conception of ‘similarity’ between activities, we expected to compute such a baseline level of HRV. Nevertheless, we agree that pre-test measurement procedures in a controlled environment (HRV upon rest and chosen activities) could be very beneficial and impact the statistical analysis in many aspects. Therefore it should definitely be suggested in future studies. 

Lines 469-478

7. Taking into the account that HRV can predict such pathological events such as sudden deaths, myocardial infarction even in young persons the revealing of extreme overload estimated by HRV may be of big use in the future. You describe it in the discussion.

We elaborated  more about general and possible outcomes of  HRV monitoring in the second paragraph of section 4.1.

Lines 450-467

Reviewer 2 Report

The goal of this preliminary study by Meina et al was to determine if sensor belts (ECG, three-axial accelerometer) could assess stress in real life settings. Investigators compared recordings and self-assessment reports of firefighters (n=12 cadets, n=14 professionals; 24-hr recordings). Accelerometer data were classified based on cluster analysis. HRV of each cluster was then compared within individuals. Results indicate positive classification of events that were (assumed?) perceived to be stressful by participants.  This paper has many strengths, most notably very thorough methods and results. I enjoyed the style of writing, which varied levels of detail and was overall pleasant to read. Strength of study design includes 24 h of monitoring, two samples of subjects, and use of both physiological and subjective measures. I believe this paper would be of interest to a variety of readers.

General comments:

  1. It should be clarified if accelerometer data alone (without HRV) can determine “stressful” events.
  2. The length of the manuscript takes away/dilutes the interesting findings. Some areas that could be condensed or some info moved to supplementary material are: 3rd paragraph of intro, data analysis/cluster analysis of motion data, discussion/limitations, conclusions.
  3. Requires minor English editing.
  4. Please spell out all acronyms upon first introduction.

Specific comments:

  1. Intro: Although in general the importance of the study is fine (i.e., chronic stress of firefighters is harmful and objective measures are important), I think results could have a stronger impact if there was a more specific framework. For example, chronic stress does not necessarily equal repeated episodes of acute stress. Could it be that objective measures of acute stress could flag need for mental health intervention or debriefing? What is the value of this technology? For example, expand on sentences 31-33, how is it “key to addressing this issue”?
  2. Intro: I would switch the order of the fourth paragraph, moving the prvious work prior to the value of machine learning. I am making the assumption that Wu et al and Comes et al do not use machine learning, and that is the novely of the current work, but this should be explicitly stated.
  3. Intro: Intro is missing hypotheses.
  4. Methods: how long after the events did subjects complete the questionnaire?
  5. Methods: Lines 248-249, hypothesis should be moved to intro.
  6. Results: do any of the questions in the questionnaire correlate with each other?
  7. Results: I enjoyed Figure 2.
  8. Results: Lines 309-310 are unclear.
  9. Results: I would take out sentence “interesting fact…” line 327. It is difficult to understand and does not seem to add value.
  10. Discussion: line 376-377, tone down word “robust”, not supported by results

Author Response

We would like to thank the reviewers very much for their effort in the reviewing process and the valuable comments and remarks. These comments and remarks have helped us for further improvement of the manuscript. Please find below our answers for specific remarks. 

Reviewer 2

  1. It should be clarified if accelerometer data alone (without HRV) can determine “stressful” events.

We added a few sentences of clarification on this topic:

“Few studies attempt to assess stress based purely on accelerometric data (Garcia-Ceja et al. 2015). Changes in movement are not necessarily correlated with variations in levels of stress. This could be true in certain occupations however firefighters tend to be physically active not only during emergency calls, but also in their free time. Physiological parameters, such as HRV, skin conductance, breathing, reflect the psychophysiological state of the body far better. Combined with the accelerometric signal these measures provide a much more complete picture of the current bodily state.”

Lines 97-113

2. The length of the manuscript takes away/dilutes the interesting findings. Some areas that could be condensed or some info moved to supplementary material are: 3rd paragraph of intro, data analysis/cluster analysis of motion data, discussion/limitations, conclusions.

According to the Reviewer’s suggestion, we moved most of the paragraph Data analysis/Cluster analysis of motion data into Supplementary Materials. We condensed the text in Discussion/Limitations and Conclusions. We hope the manuscript is now more clear and easy to read.

3. Requires minor English editing.

We performed English language proof-reading.

4. Please spell out all acronyms upon first introduction.

We corrected this shortcoming of our manuscript, and in the case of HRV indices we provide a reference to an appropriate table with explanations: 

“For  description and explanation of abbreviations of all the HRV parameters applied in this study see: Table 1.” 

Specific comments:

5. Intro: Although in general the importance of the study is fine (i.e., chronic stress of firefighters is harmful and objective measures are important), I think results could have a stronger impact if there was a more specific framework. For example, chronic stress does not necessarily equal repeated episodes of acute stress. Could it be that objective measures of acute stress could flag need for mental health intervention or debriefing? What is the value of this technology? For example, expand on sentences 31-33, how is it “key to addressing this issue”?

We created a more specific framework by modifying the following paragraph, stressing the preventive value of our technology:

“Among firefighters chronic stress is the main cause of health problems, especially heart-related diseases however, these problems do not appear overnight. It takes months or even years of psychophysiological changes to develop a chronic condition and it is usually not possible to pin-point the exact event that triggered this shift. Additionally, the amount of exposure to acute stress necessary to prompt the chronic state depends on individual characteristics. Therefore, only constant control of the pshychophysiological well-being in high-stress jobs can assure that the personnel receives the support they need exactly when they need it. This way it is possible to administer an appropriate mental health intervention or debriefing before it is too late. Real-time monitoring and on-line management of everyday stress experienced upon real-life events may be key to addressing this issue, as a preventive tool (Kunadharaju et al. 2011, Gomes et al. 2013, Lee et al. 2004, Kales et al. 2003).”

Lines 34-43

7. Intro: I would switch the order of the fourth paragraph, moving the previous work prior to the value of machine learning. I am making the assumption that Wu et al and Gomes et al do not use machine learning, and that is the novelty of the current work, but this should be explicitly stated.

Gomes and colleagues indeed did not use machine learning however, they performed a study of firefighters in ecological settings. We added a few recent examples of such research, as well as described a study of Pluntke et al., who did use artificial intelligence in their analysis of stress in firefighters, but it was set in a laboratory. The closest to our work was research done by Wu and coworkers, who applied machine learning to biological signals combining HRV and accelerometric measures collected in a natural environment. However, that study was carried out in a small sample of participants that were not firefighters. Since our research incorporates all of the aforementioned elements, we felt it was natural to present all the studies in one paragraph, and therefore decided to keep the original order of the text. However, we stated more clearly, what is the novelty of the current work:

“Little research has been done using similar methodology in the field of firefighting. A few studies attempted to assess stress levels based solely on HRV indices, both in the laboratory (Rodrigues et al. 2018) and ecological settings (Rodrigues et al. 2018, Rodrigues et al. 2018, Gomes et al. 2013). However, application of machine learning to classification of HRV-based assessment of physical, psychological and combined stress experienced by firefighters was performed only in laboratory conditions (Pluntke et al. 2019). To our best knowledge, only one study applied artificial intelligence to analysis of ecological physiological recordings that consisted of combined HRV measurements and accelerometric activity (Wu et al. 2015). The authors concluded that accelerometry significantly complemented HRV parameters, resulting in an increased stress prediction accuracy of 85,7\%. This study was performed on a small sample size of unspecified workers involved in high-stress activities. However, firefighting is an extreme occupation, resulting in working conditions not easily matched by other stressful jobs. Relying on the outcomes of previous investigations and the availability of modern, portable technology for bio-signal registration, this pilot study aims to investigate methodological means to assess the level of stress experienced by firefighters in the field.”

Lines 114-134

8. Intro: Intro is missing hypotheses.

We moved our hypotheses from the Methods section and further elaborated them:

“Relying on the outcomes of previous investigations and the availability of modern, portable technology for bio-signal registration, this pilot study aims to investigate methodological means to assess the level of stress experienced by firefighters in the field. Our hypothesis concerns the possibility to separate stressful incidents from non-incident situations by applying machine learning classification to HRV metrics collected in ecological settings from firefighters on duty. We further hypothesize that accelerometric measurements would further aid this process by providing a means to control for different types of physical activity recorded during 24-hour shifts, encompassing both the participation in emergency incidents and leisure time.  Moreover, the study attempts to classify epochs of the HRV-based biological signal according to the motion type and the level of stress experienced using various time domain and nonlinear HRV metrics in order to find the most reliable indicators of stress in the natural working environment.”

Lines 124-134

9. Methods: how long after the events did subjects complete the questionnaire?

We added a short clarification of this matter:

“The questionnaires were completed immediately after returning from the emergency call as a form of debriefing. This was purposely intended to perform the evaluation of stress in a most ecological manner possible.”

Lines 181-183

10. Methods: Lines 248-249, hypothesis should be moved to intro.

We  elaborated our hypotheses and moved them to the Introduction section (see above).

11. Results: do any of the questions in the questionnaire correlate with each other?

We performed additional statistical analysis on the questionnaire data and summarized the results in a short paragraph in the Results section:

“Several statistically significant correlations were found between the positions of the questionnaire (Spearman's rho between 0.31 and 0.79). The correlation matrix determinant equaled to 0.005, the Kaiser – Meyer – Olkin (KMO) index - to 0.6, and statistically significant results of the Bartlett’s test all indicate that our data are suitable for factor analysis. Principal component analysis with Oblimin rotation of the axes performed on the answers to the questions revealed 4 separate factors (loadings' values of the questions loading each factor are presented in brackets): (1) ‘danger‘ (Q6 = 0.885, Q4 = 0.859, Q5 = 0.770, Q7 = -0.666, Q1 = 0.539); (2) ’satisfaction’ (Q10 = 0.866, Q11 = 0.833); (3) ‘involvement’ (Q9 = 0.926, Q8 = 0.828); (4) ‘routine’ (Q3 = 0.797, Q2 = 0.736). The model explained 72.8% of variance. Reliability of the questionnaire was controlled by calculation of Cronbach's alpha, which was equal to 0.71 for the whole scale.”

Lines 277-284

12. Results: I enjoyed Figure 2.

Thank you for this comment - we are very happy to hear that!

13. Results: Lines 309-310 are unclear.

We elaborated this paragraph to make it more clear and comprehensive:

“To investigate the latter problem of  episode length influencing the HRV values, we calculated correlations between episode length and HRV values.The highest correlation Spearman rank correlation observed in our study was $.22$ ($p<.01$) for SDANN. Despite the fact, that this correlation was weak, the SDANN metric was considered possibly affected by episode length and therefore rejected from further analysis.”

Lines 311-315

14. Results: I would take out sentence “interesting fact…” line 327. It is difficult to understand and does not seem to add value.

We agree with the reviewer and therefore we removed the aforementioned sentence from the text.

15. Discussion: line 376-377, tone down word “robust”, not supported by results

We agree with the reviewer and therefore we replaced the word “robust” with “valid” in the text.

Reviewer 3 Report

no suggestions.

Author Response

Thank you very much for your review and for appreciating our work.

Round 2

Reviewer 1 Report

Dear authors! I see that the big job was done to correct the original manuscript. I am satisfied and accept the publication.